# Efficacy of a Telephone-Intervention on Caregiving Burden and Mental Health among Family Caregivers of Persons with Dementia in Malaysia: A Randomized Controlled Trial

**DOI:** 10.3390/ijerph21101354

**Published:** 2024-10-13

**Authors:** Syarifah Amirah Binti Syed Ahmad, Zarina Nahar Kabir, Marie Tyrrell, Åsa Craftman, Hashima E. Nasreen

**Affiliations:** 1Department of Community Medicine, Faculty of Medicine, International Islamic University Malaysia, Kuantan 25200, Malaysia; drnasreen@iium.edu.my; 2Department of Neurobiology, Care Sciences and Society, Karolinska Institute, 14183 Stockholm, Sweden; zarina.kabir@ki.se (Z.N.K.); marie.tyrrell@shh.se (M.T.); asa.craftman@ki.se (Å.C.); 3Nursing Institution, Sophiahemmet University, Valhallavägen, 11428 Stockholm, Sweden

**Keywords:** telephone-intervention, family caregivers, persons with dementia, burden, anxiety and depression symptoms

## Abstract

Identifying effective and accessible interventions for family caregivers of persons with dementia (PWD) is crucial as the prevalence of dementia increases in Asia. This study investigated the efficacy of a telephone-intervention on the reduction in caregiver burden, as well as depressive and anxiety symptoms among family caregivers (FCs) of PWD in Malaysia. A single-blinded randomized controlled trial was carried out with 121 FCs of PWD selected from memory or psychiatry clinics in three tertiary hospitals in Malaysia, who were randomly allocated into the intervention or control group. The intervention group received the psychoeducational intervention delivered by healthcare staff via telephone for 10 sessions over 12 weeks. The outcome of the intervention was measured by the Malay version of the Zarit Burden Interview and the Hospital Anxiety and Depression Scale at baseline and post-intervention. An intention to treat analysis shows that caregiver burden, anxiety symptoms, and psychological distress among FCs in the intervention group decreased by 7.57 units (*p* < 0.001), 2.46 units (*p* < 0.001), and 2.98 units (*p* = 0.011), respectively, at post-intervention, compared to the differences from baseline to post-intervention in the control group. Policies aimed at integrating the telephone-intervention into memory/psychiatry clinics in Malaysia may help FCs of PWD to reduce their caregiver burden and stress while caring for a family member with dementia.

## 1. Introduction

Dementia is a worldwide healthcare challenge encompassing various disorders and conditions affecting cognitive functions, particularly in persons aged 65 years and older, posing a significant burden on the person, family, and healthcare systems [1]. The World Health Organization (WHO) reported that, globally, more than 55 million people have dementia, with projections of nearly 78 million by 2030 and 139 million in 2050 [2]. Almost 9.9 million new cases of dementia are developed annually, which is one in every three seconds [3]. Approximately 60% of persons with dementia (PWD) live in low- and middle-income countries (LMICs), which includes Malaysia as well.

As longevity increases in many Asian countries due to advancements in healthcare and living standards, individuals are now experiencing longer lifespans compared to previous generations. This development is associated with a higher incidence of dementia [4]. According to the United Nations Development Programme (UNDP) [5], the Asia-Pacific region is one of the fastest aging regions in the world. Recently, the Department of Statistics, Malaysia (DoSM) [6] indicates a rise in the percentage of the population aged 65 years and above, reaching 7.2% in 2022. In 2040, it is estimated that Malaysia’s demographic landscape will have almost equal distribution between young (18.6%) and older population (14.5%) [6].

Subsequently, in Asian countries such as China, Hong Kong, Taiwan, and India, the prevalence of dementia was reported ranging from 4.2% to 7.4% [7,8,9], while in Malaysia, it was 8.5% (almost 260,000 population) in 2018 [10,11]. Family caregivers (FC) often act as informal caregivers, assisting with daily tasks, managing medications, coordinating medical appointments, and advocating for the needs of PWD [12]. The nature of these caregiving tasks can lead to social isolation, financial strains, and disruptions to both personal and family routines, contributing to a higher level of burden [13,14]. Evidence shows that as the severity of dementia increases, the caregiver’s involvement in daily tasks such as bathing, dressing, and feeding also increases, resulting in a greater caregiver burden [15,16,17,18]. Similarly, the dependency of PWD on daily life activities was found to increase caregiver burden in Malaysia [19].

In addition to the burdens related to caregivers for PWD, FCs also experience depression and anxiety symptoms, with prevalence rates of 14% to 31% and 29% to 32%, respectively [20,21,22,23]. A prospective cohort study conducted by Joling et al. [24] identified an incidence of 37% and 55% for major depressive disorder and anxiety disorder after 24 months of caregiving and 60% for comorbid depressive and anxiety disorder. In Malaysia, a study found that in the 30.5% and 20.7% of caregivers to PWD who experienced depressive and anxiety symptoms, the majority of them were women [25]. Due to traditional norms, women are often assigned caregiving roles, resulting in higher rates of mental health issues compared to male caregivers [26,27].

Cultural and social norms may influence burden and psychological distress among FCs of PWD [28,29]. For instance, in Asian countries, the belief in filial piety is deeply ingrained and widely upheld within the community, promoting positive moral values [29]. Traditionally, a significant proportion of older adults are cared for by their family members to avoid being seen as ungrateful for the sacrifices made by their elders [28]. Thus, it is important for FCs to receive education about relevant diseases, prognosis, interventions, support groups, and other community resources. This education can help improve their well-being and enhance the quality of care they provide [30,31].

Interventions for FCs through face-to-face meetings, whether in-person or in groups, were carried out in Western countries such as USA [32], Europe [33], and Spain [34], as well as in Asian countries such as Hong Kong [35], Korea [36], and Pakistan [37]. These interventions aim to provide personalized feedback and guidance on coping strategies for FCs. However, some caregivers were hesitant to share sensitive information due to privacy concerns, particularly during group discussions [33], and also reported facing challenges relating to transportation and financial constraints [38]. Additionally, internet-based interventions [39,40] and tele-rehabilitation through mobile application [41] were implemented. However, the effectiveness of these approaches may be hindered by issues related to sustainability and technical problems, such as poor internet connectivity [42].

Telephone-based interventions have evolved to provide a broader caregiver community with access to support and resources [42,43,44,45]. Telephone-based psychoeducational intervention enables caregivers to increase their knowledge about dementia, develop problem-solving skills, and facilitate social support in a cost-effective manner [46]. Although research on the effectiveness of telephone-delivered intervention in LMICs in Asia is limited [47], the existing literature suggests that the individual tailored telephone-intervention can produce favourable outcomes and could be recommended for caregivers of PWD [42,43,44,45].

Like in many Asian countries, FC burden and mental health are critical issues in Malaysia [25,48,49]. While empirical evidence of the interventions, particularly counselling and in-person and group support, has been carried out at the healthcare centre to support FCs, the demands of caregiving to PWD at home and the cost constraint makes the intervention support more challenging. Thus, this study aims to address the gaps by evaluating the effect of telephone-delivered psychoeducational interventions to reduce burden, psychological distress and anxiety, and depressive symptoms among FCs of PWD in Malaysia.

## 2. Materials and Methods

This study involved a parallel group, single-blinded randomized control trial (RCT), where FCs were recruited from the registers of PWD at the psychiatry clinic in Sultan Ahmad Shah Medical Centre (SASMEC), Kuantan; psychiatry, memory and neuromedical clinics in Hospital Tengku Ampuan Afzan (HTAA), Kuantan; and the geriatric clinic in University Kebangsaan Malaysia Medical Centre (UKMMC), Kuala Lumpur, in East and West Malaysia. Research assistants (RAs) selected the FCs and assessed their eligibility based on inclusion and exclusion criteria via telephone. The recruited study participants who met the inclusion criteria include FCs of clinically diagnosed PWD of any stage (mild, moderate or severe), aged >18 years living at home with the PWD and caring for at least 4 h/day for >6 months, able to read and understand Malay, the primary caregiver of a PWD (if there is more than one caregiver), and had access to a telephone. Participants were excluded from this study if they reported any major acute medical illness or had hearing problems, unable to communicate in Malay, or did not complete the entire questionnaire for data collection.

The sample size was calculated using the OpenEpi, version 3.01. According to Tremont et al., the improvement rate in burden and depressive symptoms as a result of an intervention was 30% [45]. The calculated sample size with a significance level of 5% and power of 80% was 49 but rounded to 50 in each group. An estimated dropout of 20% gives the required sample size of 60 in each intervention and control group, totaling 120 FCs of PWD for this study. At the initial phase of this study, 380 participants were screened for eligibility. A total of 121 participants were included, of them 60 were assigned randomly in the intervention group and the rest in the control group using a computerized randomization program (Figure 1).

A four-block randomization was performed by an independent statistician who was not involved in this study. The randomization sequence was produced using computer-generated random numbers with a block of four to ensure a 1:1 ratio of the participants in the intervention and control group. Each participant was assigned a unique ID number that was placed in a sealed opaque envelope, which contained the participant’s treatment group (intervention or control). These envelopes were bundled in groups of four, corresponding to the block size used in the randomization sequence. These envelopes were subsequently sent to the head nurse of the research, who was solely responsible for opening and distributing them to other registered nurses (RNs) and occupational therapists (OTs) for the delivery of interventions as indicated by the unique ID code on the envelopes. Allocation concealment was maintained by concealing this allocation sequence from those who were involved with this study.

Participants assigned in the intervention group received the psychoeducational intervention delivered by the healthcare staff via telephone over 10 sessions in 12 weeks (Figure 1). Before the intervention, the healthcare staff, consisting of five RNs and two OTs, attended a three-day online workshop on FC enrolment, psychoeducational intervention modules, and its implementation strategies. The training was provided by the researchers and clinicians from Karolinska Institute, focusing on active learning by encouraging questions and utilizing participant role-playing. Subsequently, a one-day follow-up training session for the healthcare staff providing the intervention was conducted in Malay to sharpen knowledge before delivering the intervention.

The intervention booklet inspired by the WHO’s iSupport training and support manual for carers of PWD [50] was posted to the caregivers allocated under the intervention group by the head nurse once the baseline assessment was completed. The WHO recommends the iSupport program for caregivers of PWD across all the 194 member states. The iSupport program offers various learning opportunities, training modules, support groups [2], and access to web-based resources. The healthcare staff, trained on implementing the intervention, started the initial call, and subsequent follow-up calls were scheduled based on the accessibility and availability of both the healthcare staff and the caregivers. Each call lasted for around 30 min, and each FC was assigned a specific RN or OT for the whole duration of the intervention. Every session provided by the healthcare staff was ended with a task for the next session. Participants noted the assigned tasks, tips, and any issues in their caregiver booklet to discuss in the next session. The study protocol, which outlined the intervention contents and tasks, are summarized in Table 1.

The participants completed the baseline survey from August 2022 to February 2023 on caregiver burden, and psychological distress involving anxiety and depression symptoms through a structured questionnaire administered by the trained RA via telephone. The trained RA who assessed the participants was blinded to the treatment group (whether intervention or control). However, blinding was not extended to the nurses responsible for delivering the intervention, nor to the statistician who had knowledge of which participant’s ID received the treatment. Each FC was informed about the research project, psychoeducational intervention, risks, and benefits to participate in this study. Only those FCs who gave their voluntary consent were enrolled. In addition, participants in both control and intervention groups received the usual care and information that was available at the hospitals or clinics. A post-intervention assessment was carried out at week 14 to assess the outcome measures, including caregiver burden and depressive and anxiety symptoms. Background variables in this study included socioeconomic characteristics such as the family caregiver’s age, sex, education (primary, secondary, or tertiary); occupation (employed employee, homemaker/unemployed, retired caregiver); monthly household income; caregiving information (length of caregiving, hours of caregiving per day, if the caregiving was shared by other family members, number of family members involved in shared caregiving, caregiver’s relationship with PWD); caregivers’ perceived social support (family, friends, and significant others’ support); and PWD’s demographic information such as age, sex, and the ability to selfcare. Caregivers’ perceived social support was assessed using a validated Malay Version of Multidimensional Scale of Perceived Social Support (MSPSS), which consisted of 12 items: family support (4 items), friends’ support (4 items), and significant others’ support (4 items), scored from 1 (very strongly disagree) to 7 (very strongly agree). The total score was between 0 and 84, in which a higher score indicated higher social support [51]. The internal consistency of the caregiver’s social support for the whole scale and the three subscales were found to have a Cronbach’s alpha between 0.91 and 0.93.

The main end points for the caregivers of PWDs were changes in caregiver burden, psychological distress, and anxiety and depressive symptoms between baseline and post-intervention assessment. The validated Malay version of the Zarit Burden Interview (ZBI) was used to measure caregivers’ burden, which consists of 22 items, scored from 0 to 4 on each item [52]. The total score was between 0 and 88, where a higher score indicated a higher level of burden. The reliability and validity of the Malay version of the ZBI (MZBI) was verified, whereby a score of 22 represented the optimum cut-off point for 70.8% sensitivity and 69.2% specificity [52]. The internal consistency of the MZBI in the original study was assessed using Cronbach’s alpha, which yielded a good internal consistency of 0.90. In our study, the Cronbach’s alpha of MZBI was 0.92 at baseline and 0.93 at post-intervention. The validated Malay version of the Hospital Anxiety and Depression Scale (HADS) was used to measure psychological distress [53], which consisted of 14 items, scored from 0 to 3 on each item. The HADS questionnaire consisted of two subscales: anxiety (HADS-A) and depressive (HADS-D) symptoms. Each subscale comprised seven items with the total score between 0 and 21, where a higher score indicated higher psychological distress, anxiety, and depressive symptoms. The scale was developed as a respondent’s verbal response tool [53] and demonstrated good reliability with Cronbach’s alpha 0.88, 0.84, 0.78 at baseline and 0.88, 0.83, 0.82 at post-intervention for psychological distress, anxiety, and depressive symptoms.

The data were analyzed using IBM SPSS Statistics version 26.0. Bivariate analyses, i.e., Chi-square or Fisher exact tests, independent t-tests, and Mann–Whitney U tests, were conducted to compare the participants’ baseline characteristics between intervention and control groups. Mixed ANOVA was used to examine the net gain effect of intervention on the outcome variables. An intention to treat (ITT) analysis, using a linear mixed model for repeated measures and adjusted for all possible associated factors, was conducted to evaluate the independent effect of intervention on the outcome variables. Before conducting the ITT analysis, independent t-tests identified the possible baseline characteristics associated with outcome measures. A *p*-value of <0.05 was considered for statistical significance. An ITT analysis included all randomized patients (N = 121) regardless of subsequent withdrawal from the protocol [54].

This study was approved by the Malaysia Medical Research and Ethics Committee (NMRR ID-22-00137-BUY), IIUM Research and Ethics Committee (IREC 2022-007), Department of Psychiatry and Memory, Hospital Tengku Ampuan Afzan (HTAA) (00137-BUY (2)), Department of Psychiatry, Sultan Ahmad Shah Medical Centre (IIUM/413/013/14/11/2/IISR22-09), and Research Ethics Committee of University Kebangsaan Malaysia (UKMPPI/11/8/JEP-2022-328). Caregivers provided informed consent after receiving comprehensive information about this study’s purpose, procedures, risks, and benefits, with the assurance of voluntary participation and the right to withdraw from this study at any time without repercussions. The respondent’s identities were kept confidential, the data were collected anonymously, and anonymity was maintained in publishing the data.

## 3. Results

### 3.1. Response Rate and Adherence to the Intervention

Among the 121 participants, 16 (13%) dropped out from this study, of which 11 (18%) were from the intervention group and 5 (8%) were from the control group (Figure 1). The reasons for drop out were time constraint (38%), deceased PWD (25%), participants were unreachable (19%), participants were unwilling to continue in this study (12%), and one PWD went missing (6%) during the study time (Figure 1).

### 3.2. Baseline Characteristics of the Participants

Baseline socioeconomic and caregiving characteristics of the FCs and demographic characteristics of the PWD were similar between the intervention and control groups, except for the participant’s sex and relationship between dyads. About 70% of the FCs were female and 63% were PWD’s own children with significantly higher proportions in the intervention group (*p* = 0.004 and *p* = 0.041, respectively). The mean age of the FCs was 51.6 (12.7). Most of them were Muslims, married, had a tertiary level of education, and employed with a median monthly household income of RM 4000 (USD 1 = RM 4.7). Moreover, the two groups were similar in their outcome measures at baseline (Table 2). FCs of PWD who dropped out from this study were wealthier (*p* = 0.031) and had higher years of schooling (*p* = 0.015). No significant differences were observed on the other baseline characteristics between participants who dropped out and who remained in this study.

### 3.3. Intervention Effects

Mixed ANOVA (Table 3, Figure 2) showed no significant changes in mean ZBI, HADS-A, HADS-D, and HADS-total scores from baseline to post-intervention time points and between the intervention and control groups. However, the interaction between receiving the intervention and time (Figure 2) indicates significant negative net gain scores for all outcome variables except for the HADS-D (Table 3). Independent t-tests revealed that if caregivers perceived themselves as non-Muslim, had lower monthly household income, and had lower social support, they reported higher mean score on caregiver burden, depressive and anxiety symptoms, and psychological distress. Additionally, if the PWD were not able to selfcare, the caregivers reported higher caregiver burden while unmarried caregivers exhibited higher mean scores of burden, anxiety, and psychological distress.

Table 4 shows the results of the intention to treat analysis using a linear mixed model for repeated measures and examined the independent effect of intervention on the outcome variables over time while controlling for the possible associated factors, such as caregiver’s religion, monthly household income, marital status, social support, and the PWD’s ability to selfcare. Except for HADS-D, a significant interaction between receiving the intervention and time was observed on all outcome measures. Caregiver burden among the participants in the intervention group decreased by 7.57 units on ZBI at T1 (post-intervention) (*p* < 0.001) compared to the difference between T1 and T0 in the control group. In terms of psychological outcome, anxiety symptoms and psychological distress scores among participants in the intervention group decreased over time by 2.46 (*p* < 0.001) on HADS-A and 2.98 (*p* = 0.011) on HADS-total compared to the differences from baseline to post-intervention in the control group.

## 4. Discussion

The present study evaluated the effectiveness of a telephone-delivered psychoeducational intervention in reducing FCs’ burden, psychological distress, anxiety, and depressive symptoms. The results revealed significant improvements in caregiver burden, psychological distress, and anxiety in the intervention group compared to the control group. However, no significant change was observed in depressive symptoms. These outcomes align with previous studies on psychoeducational interventions, which highlighted the benefits of educational sessions and problem-solving strategies in reducing caregiver burden and anxiety [46,56,57]. Psychoeducational interventions were found to be effective in alleviating burden and anxiety, while acceptance and commitment therapy were considered more effective for addressing depression [58].

The results suggest that telephone-based intervention significantly reduced the caregiver burden of PWD compared to the standard care, which is in line with similar results as shown by Kwok et al. [44]. In contrast, Tremont et al. found no intervention effects on caregivers’ burden after 6 months of intervention compared to our study, which was conducted in a 3-month period [45]. Tremont et al. suggested a longer intervention duration for at least 12 months to observe a substantial reduction in caregiver’s burden [45]. However, the observed differences shows that the duration of the intervention alone may not fully account for the differences in the outcomes but might be attributed to variations in the specific contents of the interventions. Additionally, Davis et al. showed a reduction in caregiver burden on telephone psychoeducation and skill training [59]. However, like Davis et al. compared with in-home intervention, telephone intervention may take longer time to reduce burden and contribute to a higher attrition rate compared to in-home psychoeducation intervention. In-home intervention may be beneficial in terms of personal interaction but may not be feasible to all FCs of PWD. Conversely, continued research by Chodosh et al. indicates no difference in caregiver burden between the telephone approach and in-home plus telephone intervention [60]. It shows that telephone intervention can only be as effective as home visits to achieve the desired outcome variables. Since telephone-based intervention can be seen as a doable method to support FCs of PWD, it requires reliable strategies to enhance engagement and retention.

Significant improvements were also observed in FCs’ psychological distress and anxiety symptoms. This observation is consistent with an intervention study conducted by Hattink et al., which found significant differences in psychological distress through website portal and telephone consultation after 2 months of intervention compared to standard care [61]. However, there was no improvement in anxiety among FCs of PWD [60]. In contrast to expectations, no statistically significant effect of intervention on depressive symptoms were found in the current study. This outcome is consistent with the results shown by Soylemez et al., where intervention conducted through home visits followed by telephone calls showed no significant reduction in depressive symptoms [62]. Similarly, Martin-Carrasco et al. found no significant reduction in depressive symptoms as a result of an educational intervention program based on coping with caregiving (CWC) [63]. A recent intervention study in low-income countries, such as India [64], also shows that online intervention training using iSupport did not find significant results in depressive symptoms. However, this study showed a high attrition rate (approximately 60%) in the intervention study, which limits the ability to draw reliable conclusions on the effect of intervention. Although some intervention studies did show the effectiveness of telephone interventions in reducing depressive symptoms [43,45,65], several factors could explain the discrepancies with our results. First, the intervention period in our study was spanned over a three-month period, whereas previous studies showing improvement in depressive symptoms had longer interventions lasting at 6 months [45], 15 months [64], and 18 months [43]. Additionally, the mean depression score among participants was 5.5 (SD = 4.2), showing the mean score near the lower end of the scale, which indicates that the participants in our study had initially lower levels of depressive symptoms compared to burden and anxiety, reducing the likelihood of detecting significant changes in depressive symptoms.

The high level of the participant response rate with the current intervention highlighted the caregiver’s evident need for support. However, discontinuing the intervention after this study could lead to a potential reduction in the intervention effect [66]. Further investigation is needed to maximize the benefit of the intervention. For instance, Losada et al. suggested adding a few booster sessions as a strategy to maintain the sustainable effect of the interventions [34]. This study also faced limitations, particularly concerning attrition rates among higher income FCs and those with more years of schooling. This subgroup might potentially be influenced by their access to other effective complementary treatment for themselves and their family members with dementia. FCs with advanced levels of schooling may contribute to higher-level reasoning and problem-solving skills that result in better access to information and multiple options of care and treatment, utilizing the given information in a way to maximize its benefit [67], which is outside our study’s scope. In addition, time constraints were found as the main reason for attrition, suggesting higher income FCs possibly prioritized work commitments. Another limitation is that the PWD’s stages of severity of dementia were not investigated in this study. The needs of the FC may vary depending on the stage of the person’s dementia, from moderate to severe; this potential variation, in turn, impacts specific needs, coping strategies, and challenges [68].

One of the significant challenges encountered by the healthcare staff during the telephone-intervention of caregivers with the PWD was the necessity to reschedule the intervention sessions due to time limitations. The process of rescheduling intervention sessions may have introduced anxiety and stress for both the FCs and the healthcare staff, particularly when arranging suitable call times. Consequently, we offered flexibility to allow caregivers and healthcare staff to schedule calls at their convenience. Despite these limitations, this study design was a single-blinded randomized control trial, which is an appropriate design to measure the efficacy of an intervention. Participants’ allocation concealment and the RA’s blindness to treatment groups helped minimize the selection and information bias in this study. We also tried to minimize the attrition bias by using both the per-protocol (mixed ANOVA) and intention to treat (mixed model) analyses that shows almost similar results. Additionally, locally validated instruments were used to measure the outcomes. The intervention also offered flexibility in terms of the caregiver’s availability and discussion topics, which were accessible and convenient for busy caregivers who were employed, homebound, or resided in rural areas. Tremont et al. suggested that telephone-based intervention also helps in reducing costs by eliminating the need for physical space and travel expenses [45].

## 5. Conclusions

This study underscores the potential of telephone-delivered psychoeducational interventions in reducing caregiver burden, psychological distress, and anxiety among FCs of PWD. While significant improvements were observed in these areas, this study did not find significant effects on depressive symptoms. This outcome highlights the need for further research to examine the most effective strategy and appropriate treatment duration to address depression effectively. Overall, the support of FCs using telephone-based intervention could be one of the solutions to help policy makers implement the intervention model in geriatric and psychiatric clinics or hospitals to improve the burden, anxiety, and psychological distress of FCs to PWD in the community in Malaysia. The policymakers may consider allocating funding for training healthcare professionals to improve the quality of care for PWD.

## Figures and Tables

**Figure 1 ijerph-21-01354-f001:**
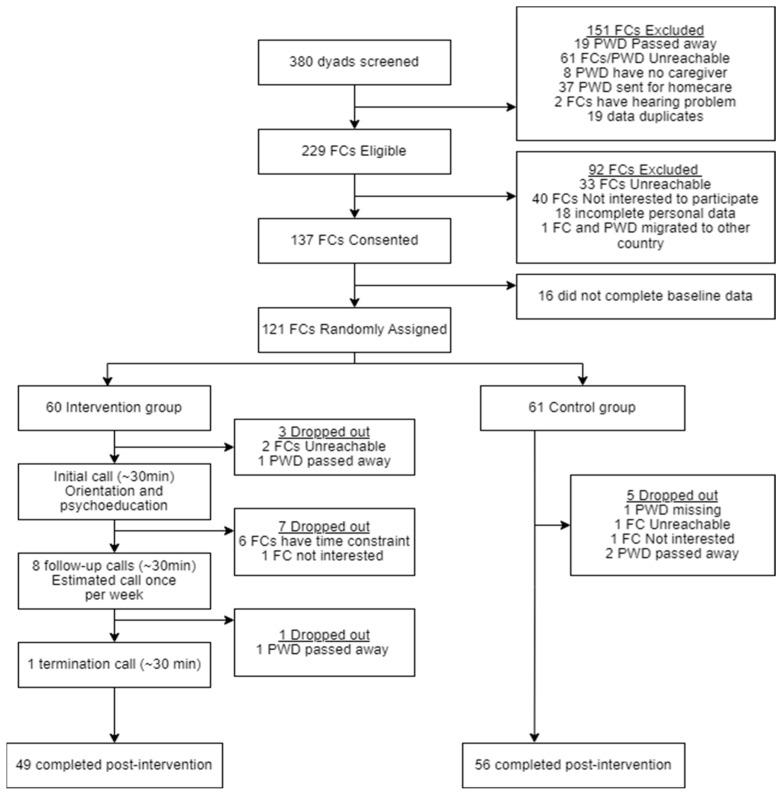
Flowchart of participant enrolment.

**Figure 2 ijerph-21-01354-f002:**
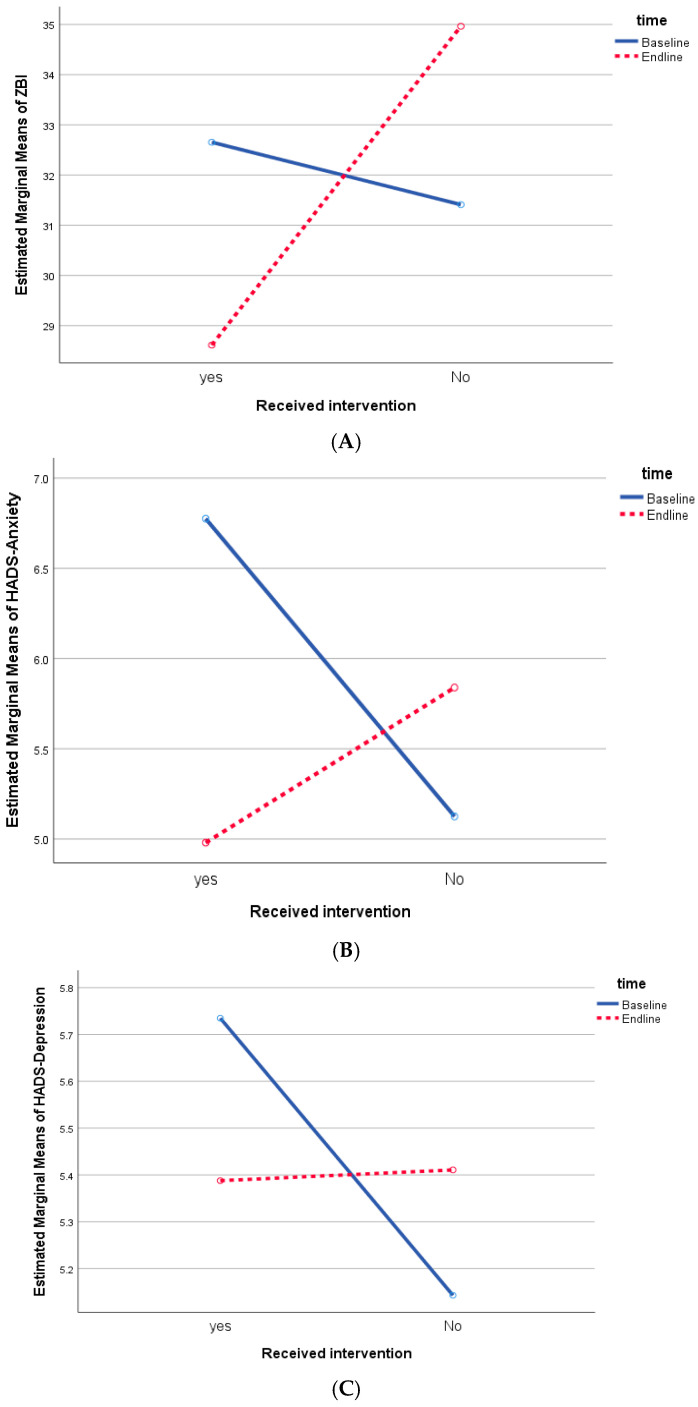
Interaction between time and intervention on outcome variables: (**A**) caregiving burden (ZBI); (**B**) anxiety symptoms (HADS-A); (**C**) depressive symptoms (HADS-D); and (**D**) psychological distress (HADS-Total).

**Table 1 ijerph-21-01354-t001:** Intervention protocol.

Sessions	Topics	Contents	Tasks for Next Session
Session 1	Introduction	Rapport building, risk appraisal, challenges, and positive aspects of caregiving	Knowledge on dementia, its diagnosis, and identification on the possible challenges
Session 2	Introduction to dementia	Information about dementia: types, stages, and challenges	Role of family caregiver and how to perceive the relationship to PWD
Session 3	Being a family caregiver I	Support caregivers to understand their role as caregivers	Changes observed in caregiver’s life
Session 4	Being a family caregiver II	Coach caregivers on utilization of accessible tools and strategies for their recreation	Issues on mental health and strategies they overcome
Session 5	Mental strategies and mind-set	Assist caregivers about mental strategies and ways of thinking	Needs, issues, and challenges concerning their caregiving role
Session 6	Free choice of topic	Topic that caregivers felt was more important to discuss	Activities performed by the caregivers: want to do, do not want, and able to do
Session 7	Activities	Support caregivers to utilize activities strategically	Issues on behavioural and emotional changes in PWD
Session 8	Behavioural and psychological symptoms in dementia (BPSD)	Help caregivers to manage BPSD	Issues concerning nutrition (food, eating, weight loss) and/or sleep disturbances
Session 9	Eat and sleep	Help caregivers to identify appropriate strategies to solve problems related to eating and sleeping	Resources that family caregivers need
Session 10	Continued resources	Encourage caregivers to utilize the existing support system in the family, community, and the government system	-

**Table 2 ijerph-21-01354-t002:** Baseline characteristics of family caregivers and persons with dementia (PWD).

	Total Sample	Intervention	Control	*p*-Value
	*n* = 121	*n* = 60	*n* = 61	
Family Caregivers’ Socioeconomic Characteristics				
Age (years), Mean (SD)	51.6 (12.7)	50.1 (12.4)	53.1 (12.9)	0.185
Sex (%)				
Male	30.6	18.3	42.6	0.004
Female	69.4	81.7	57.4	
Religion (%)				
Muslim	66.9	75	59	0.057
Hindu/Buddhist/Christian	33.1	25	41.7	
Education (%)				
Primary	13.2	15.0	11.5	0.365
Secondary	39.7	33.3	45.9	
Tertiary	47.1	51.7	42.6	
Years of schooling, Mean (SD)	12.9 (3.4)	12.9 (3.5)	13.0 (3.3)	0.935
Marital status (%)				
Unmarried	18.2	18.3	18	0.553
Married	73.6	71.7	75.4	
Divorced/widowed	8.2	10.0	6.6	
Occupation (%)				
Employed	54.5	45.0	63.9	0.111
Homemaker/unemployed	35.5	43.3	27.9	
Retired	10.0	11.7	8.2	
Monthly HH income (RM), Median (IQR)	4000 (69,500)	4000 (69,250)	4000 (29,500)	0.977
Monthly HH income (RM), (%)				
B40 (RM4580 and below)	56.2	60	52.5	0.667
M40 (RM4581-RM10959)	35.5	31.7	39.3	
T20 (RM10960 and above)	8.3	8.3	8.2	
Caregiving Information				
Length of caregiving (months), Mean (SD)	47.9 (42.8)	40.7 (34.3)	55.1 (49.0)	0.064
Hours of caregiving/day, Mean (SD)	18.6 (6.9)	18.8 (6.9)	18.4 (7.1)	0.8
Shared caregiving by other family members (%)	60.3	56.7	63.9	0.414
Number of persons involved in shared caregiving, Mean (SD)	1.3 (1.5)	1.4 (1.7)	1.3 (1.3)	0.89
Relationship with person with dementia (%)				
Spouse	27.3	21.7	32.8	0.041
Adult child	62.8	73.3	52.5	
In-laws	9.9	5.0	14.8	
Social Support, Mean (SD)				
Social support (total)	59.3 (17.1)	58.6 (16.8)	59.9 (17.6)	0.682
Family support	21.2 (6.4)	21.3 (6.0)	21.5 (6.7)	0.891
Friend support	16.1 (7.3)	16.0 (7.5)	16.3 (7.2)	0.844
Significant other support	21.7 (6.5)	21.3 (6.9)	22.2 (6.1)	0.466
Zarit Burden Interview (ZBI), Mean (SD)	32.9 (18.4)	34.0 (18.5)	31.8 (18.2)	0.510
HADS, Mean (SD)				
HADS-total score	11.7 (8.2)	12.6 (8.1)	10.8 (8.2)	0.227
HADS-D score	5.5 (4.2)	5.7 (4.3)	5.4 (4.2)	0.693
HADS-A score	6.2 (4.6)	6.9 (4.3)	5.5 (4.9)	0.090
PWD’s Demographic Information				
Age (year), Mean (SD)	75.2 (10.1)	74.8 (10.2)	75.6 (9.9)	0.661
Sex (%)				
Male	37.2	38.3	36.1	0.796
Female	62.8	61.7	63.9	
Able to selfcare (%)	56.2	50	62.3	0.173

**Table 3 ijerph-21-01354-t003:** Mean scores and gain scores on ZBI and HADS by intervention groups (N = 105).

		Mean Score (SD)		Gain Score	*p*-Value	Net Gain Score (95% CI)	*p*-Value	Effect Size (Partial η^2^)
		T0 (Baseline)	T1 (Post-Intervention)					
ZBI	Intervention	32.65 (19.00)	28.61 (18.14)	−4.04	0.787	−7.59 (−11.16–−4.03)	<0.001	0.148
	Control	31.41 (18.50)	34.96 (19.67)	3.55				
	Difference	1.24	−6.35					
	*p*-Value	0.992						
HADS-A	Intervention	6.78 (4.18)	4.98 (3.78)	−1.80	0.111	−2.51 (−3.84–−1.18)	<0.001	0.119
	Control	5.13 (4.76)	5.84 (4.94)	0.71				
	Difference	1.65	−0.86					
	*p*-Value	0.201						
HADS-D	Intervention	5.73 (4.42)	5.39 (4.89)	−0.35	0.909	−0.62 (−1.97–0.75)	0.373	0.008
	Control	5.14 (4.03)	5.41 (4.25)	0.27				
	Difference	0.59	−0.02					
	*p*-Value	0.956						
HADS-Total: Psychological Distress	Intervention	12.51 (8.08)	10.37 (7.97)	−2.14	0.319	−3.13 (−5.42–−0.83)	0.008	0.065
	Control	10.27 (8.10)	11.25 (8.25)	0.98				
	Difference	2.24	−0.88					
	*p*-Value	0.426						

Gain score = post-intervention score − baseline score; Net gain score = gain score (intervention) − gain score (control); partial η^2^: small effect (η^2^ = 0.01 to 0.06), moderate effect (η^2^ > 0.06 to 0.14), and large effect (η^2^ > 0.14) (Richardson, 2011) [55].

**Table 4 ijerph-21-01354-t004:** Linear mixed model for repeated measures showing the effects of intervention on caregiving burden and psychological distress of family caregivers to persons with dementia (N = 121).

	Unadjusted				Adjusted			
	®	Std. Error	95% CI	*p*-Value	®	Std. Error	95% CI	*p*-Value
ZBI								
IG (ref: CG)	2.18	3.37	−4.49–8.51	0.519	1.38	3.11	−4.78–7.54	0.657
T1 (ref: T0)	3.50	1.22	1.08–5.93	0.005	3.47	1.22	1.05–5.89	0.005
IG × T1 (ref: CG × T0)	−7.71	1.79	−11.25–−4.17	<0.001	−7.57	1.78	−11.11–−4.03	<0.001
HADS-A								
IG (ref: CG)	1.54	0.82	−0.09–3.17	0.063	1.44	0.76	−0.07–2.95	0.061
T1 (ref: T0)	0.64	0.45	−0.27–1.54	0.167	0.62	0.46	−0.29–1.52	0.179
IG × T1 (ref: CG × T0)	−2.48	0.67	−3.80–−1.16	<0.001	−2.46	0.67	−3.78–−1.14	<0.001
HADS-D								
IG (ref: CG)	0.26	0.80	−1.32–1.84	0.749	0.36	0.75	−1.13–1.86	0.630
T1 (ref: T0)	0.18	0.46	−0.75–1.10	0.704	0.16	0.47	−0.77–1.08	0.738
IG × T1 (ref: CG × T0)	−0.51	0.67	−1.86–0.84	0.456	−0.53	0.68	−1.88–0.82	0.439
HADS-Total: Psychological Distress								
IG (ref: CG)	1.80	1.48	−1.13–4.72	0.227	1.74	1.35	−0.94–4.41	0.201
T1 (ref: T0)	0.84	0.78	−0.73–2.40	0.291	0.79	0.78	−0.77–2.35	0.317
IG × T1 (ref: CG × T0)	−3.01	1.14	−5.29–−0.73	0.010	−2.98	1.14	−5.26–−0.70	0.011

IG: intervention group; CG: control group; T0: baseline; and T1: post-intervention. Models were adjusted for FC’s religion, monthly household income, marital status, social support, and PWD’s ability to selfcare.

## Data Availability

The datasets generated and analyzed during the current study are not publicly available due to confidentiality issues but are available from the principal investigator upon reasonable request.

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
