# Peer review of "Efficacy of a Telephone-Intervention on Caregiving Burden and Mental Health among Family Caregivers of Persons with Dementia in Malaysia: A Randomized Controlled Trial"

_ijerph, 2024, doi:10.3390/ijerph21101354_

Round 1
Reviewer 1 Report
Comments and Suggestions for Authors
In Table 1, Hours of caregiving/day, mean is shown 18 hours. Considering that about half of the caregivers are employed in both groups, it does not seem correct.
In the results section, no interpretation or conclusion should be made, such as “This indicates that caregiver burden, anxiety symptoms and psychological distress were reduced after receiving the intervention over time” in line 254-255.
Author Response
Please see the attachement.

Reviewer 2 Report
Comments and Suggestions for Authors
1. It is an interesting, well-controlled study. Please revise Line 110 as the sentence is not clear. Also, recheck the list of references since most of them don't have page numbers.
2. The WHO’s iSupport training for carers of PWD is in English, and the training was conducted in Malay. Have the authors checked the validity of the translated version?
3. In the limitation paragraph of the discussion, it was mentioned that “The study also faced limitations, particularly concerning attrition 345 rates among higher-income FCs and those with more years of schooling”. This information was not shared in the result section. For instance, income was referred to as the median monthly income of 4000 RM without classifying it as low, medium, and high. These points need to be presented in the result section before being discussed or even mentioned in the limitation of the study.
Comments on the Quality of English LanguageMinor checking. Some sentences are hard to follow because they are too long. Try to shorten these sentences
Reviewer 3 Report
Comments and Suggestions for Authors
Congratulations on a very good work. A very interesting research concept.
In your conclusions you write that your results can be used to create strategies for decision makers. I think it is worth seeing if you are able to assess the cost-effectiveness of the proposed intervention.
Author Response
Please see the attachement.

Reviewer 4 Report
Comments and Suggestions for Authors
Very good article agree to break it down after relating to my comments:
Summary of Recommendations:
Clarify the randomization process and power calculation.
Expand on the specifics of the intervention and how bias was controlled.
Address the dropout rates and any potential impact on results.
Improve tables and figures for clarity.
Provide a deeper discussion on depressive symptoms and compare results with other LMIC studies.
Strengthen the conclusion with policy implications.
Comments on the Quality of English Language
see above
